behaviour, biomechanics, ecology

biotelemetry, social behaviour, olive baboons, consensus costs, intermittent locomotion

**Authors for correspondence:**
Roi Harel
e-mail: rharel@ab.mpg.de
Margaret C. Crofoot
e-mail: mcrofoot@ab.mpg.de

# Locomotor compromises maintain group cohesion in baboon troops on the move

Roi Harel[1,2,3], J. Carter Loftus[1,2,3] and Margaret C. Crofoot[1,2,3,4]

[1]Department for the Ecology of Animal Societies, Max Planck Institute of Animal Behavior, Bücklestraße 5, Konstanz D-78467, Germany
[2]Department of Biology, University of Konstanz, 78457 Konstanz, Germany
[3]Department of Anthropology, University of California, Davis, CA 95616, USA
[4]Center for the Advanced Study of Collective Behavior, University of Konstanz, 78464 Konstanz, Germany

RH, 0000-0002-9733-8643; JCL, 0000-0002-0723-9159; MCC, 0000-0002-0056-7950

When members of a group differ in locomotor capacity, coordinating collective movement poses a challenge: some individuals may have to move faster (or slower) than their preferred speed to remain together. Such compromises have energetic repercussions, yet research in collective behaviour has largely neglected locomotor consensus costs. Here, we integrate high-resolution tracking of wild baboon locomotion and movement with simulations to demonstrate that size-based variation in locomotor capacity poses an obstacle to the collective movement. While all baboons modulate their gait and move-pause dynamics during collective movement, the costs of maintaining cohesion are disproportionately borne by smaller group members. Although consensus costs are not distributed equally, all group-mates do make locomotor compromises, suggesting a shared decision-making process drives the pace of collective movement in this highly despotic species. These results highlight the importance of considering how social dynamics and locomotor capacity interact to shape the movement ecology of group-living species.

## 1. Introduction

Group-living animals incur consensus costs when they compromise their own preferred course of action to remain in contact with other members of their group [1]. When group members vary in their physical characteristics (e.g. body size), consensus costs may be particularly high, as physiological differences can introduce significant conflicts of interest among group-mates. Differences in locomotor capacity—the ability of an organism to move through its environment—may pose particularly severe challenges to behavioural coordination in heterogeneous groups. Locomotor capacity, which is dependent on a range of morphological features including body weight and limb length, affects the energetic costs of movement and therefore serves as a major driver of movement decisions [2,3]. Studies of several species of terrestrial animals reveal that individuals have a preferred travel speed [4], which is hypothesized to maximize energy efficiency [5]. Because physical characteristics such as limb length and body mass shape preferred travel speeds [6], variation among individuals in body size will lead to differences in optimal stride frequencies and travel speeds within groups. How do groups maintain cohesion during collective movement when faced with such inter-individual differences?

The locomotor choices that individuals make with respect to stride frequency and length have important effects on their energetic cost of transport (bipeds [7]; quadrupeds [8,9]). Despite the obvious potential for differences in preferred travel speed and stride frequency to introduce behavioural and energetic conflicts of interest when individuals move together as a group, our understanding of the impact of locomotor capacity on collective movement is limited [10,11]. In order to maintain group cohesion, individuals are expected to alter their patterns of movement to cope with differences in movement capacity [12]. Indeed, smaller individuals in groups of pigeons (*Columba livia*)

fly faster than their preferred travel speed whereas larger individuals fly slower than preferred, allowing them to remain together while on the move [11]. For terrestrial animals, however, individuals can adjust their movement patterns not only by changing their travel speed, but also by rapidly switching between moving and pausing phases (i.e. intermittent locomotion).

Cohesion can thus be maintained in two ways: individuals with higher locomotor capacity can slow down or pause to allow other group members to catch up, or individuals with lower locomotor capacity can travel faster or pause less frequently to keep up with their group-mates. In either case, some group members pay a cost. Faster animals who slow down or pause and wait pay an opportunity cost because they commit additional time to transit that could have otherwise been devoted to other activities such as feeding. Individuals who speed up to remain with their group, or who take fewer breaks during travel, increase their energetic cost of locomotion and may miss opportunities to forage 'on the go'. On the other hand, if group members fail to coordinate, the resulting increase in group spread jeopardizes the benefits realized by maintaining cohesion, which include increased information transfer [13,14], reduced predation risk [15], enhanced foraging efficiency [16] and more social opportunities. Because these benefits are not experienced equally by each group member, some individuals may be willing to compromise to a greater degree to remain with their group-mates.

To test how members of heterogeneous groups maintain cohesion during collective movement, we tracked a troop of wild olive baboons (*Papio anubis*). Olive baboons live in groups of up to 150 individuals that travel together throughout the day in search of resources, and sleep together at night. Females are philopatric and gain substantial fitness benefits from living with female kin and from the social bonds that they form with these kin [17]. Residing in stable groups confers a range of benefits above and beyond the decreased predation risk, information exchange and foraging benefits that arise from aggregating with conspecifics [18]. In addition to dominance hierarchies and nepotistic kin relations, strongly differentiated social bonds structure baboon societies, and these 'friendships' provide significant fitness benefits for group members [19]. However, this social organization also presents challenges to group cohesion: because they are sexually dimorphic and live in mixed-age groups, baboon troops exhibit large within-group variation in body size [20] and, thus, locomotor capacity.

We first tested whether differences in body size translated into differences in stride frequency and daily travel distances, as well as dynamic body acceleration which is a proxy for energetic expenditure [21]. We then investigated how fine-scale movements preserved group cohesion, and, in doing so, identified decision rules that could generate the observed patterns of spacing, taking individuals' body size and relative position in the group into account. Baboons group-mates do not benefit equally from their membership in their troop [19,22], and thus we hypothesized that individuals who had more to gain from group membership would be willing to incur additional locomotor costs to remain with the group. Because smaller and younger individuals are typically more vulnerable to predators [23], and are expected to be less experienced [24] we predicted that smaller baboons would be more sensitive to their spatial positioning, make larger behavioural

compromises, and bear more of the costs of maintaining group cohesion, compared to larger group members.

## 2. Methods

### (a) Data collection
We fit GPS collars with integrated tri-axial accelerometers to 25 wild olive baboons (*P. anubis*) belonging to a single group at the Mpala Research Centre in Laikipia, Kenya (figure 1). Collar units recorded location estimates continuously at a 1 Hz sampling interval and tri-axial acceleration data at 12 Hz during daylight hours (6–18 h) from 1 August to 2 September 2012. While individuals were chemically immobilized and being fit with telemetry collars (see [25] for details on capture methodology), the length of each individual's front leg was measured (dorsal most point of the scapula to the carpus; hereafter referred to as 'leg length,' which is a primary anatomical driver of locomotor costs in terrestrial animals [26,27] (see electronic supplementary material, table S1 for morphological metrics). Collared individuals consisted of 54% (25/46) of all group members (and 80% of the adults and subadults; 23/29), including 13 adults, 10 subadults and two juveniles. Two of the adults were removed from the analyses due to missing body measurements or tag malfunction that resulted in irregular acceleration data sampling, resulting in 255 whole days on 23 individuals.

### (b) Inferring behaviour at the individual and group level
Individual's activity state (moving or non-moving) was inferred with a support vector machine following [28] (more details in the electronic supplementary material). To test how baboons maintain cohesion when moving as a group, we identified periods of collective movement. Group activity state was classified into two categories, stationary and non-stationary, based on changes in the displacement of the group centroid. Group travel bouts were classified using a change point detection algorithm [29] on the centroid displacement speed.

### (c) The influence of body size on locomotion
To determine if variation in body size translates to differences in preferred gait characteristics, we tested for a relationship between an individual's leg length and its characteristic stride frequency. We estimated stride frequency based on the timing of heave-axis (i.e. baboons' dorsal–ventral axis) peaks in the accelerometry data [9] (figure 1*b*) after applying a Hampel filter to the acceleration data to remove spikes [9] that were likely caused by physical strikes to the collar. For each individual, we estimated a characteristic stride frequency by measuring the average stride frequency while the focal individual was moving, but the rest of its group-mates were stationary (i.e. independent movement). Thus, the characteristic stride frequency provides an individual-specific reference value that represents the stride frequency each individual chooses, independent of the need to maintain group cohesion. We then tested for a correlation between individuals' characteristic stride frequencies and their leg lengths during both independent and group movement. Because collars were fitted in the same manner to animals and were weighing 1–3.4% of an individual's body weight, we do not expect that they influenced the gait of larger versus smaller individuals differently.

To assess how movement patterns vary with body size, we calculated the daily travel distance and daily maximum displacement from the sleeping site of each group member, as well as of the group's centroid. Daily travel distance is a widely used measure of animal movement but is strongly affected by sampling frequency [30,31]. For this reason, and to avoid the

*Proc. R. Soc. B* **288**: 20210839

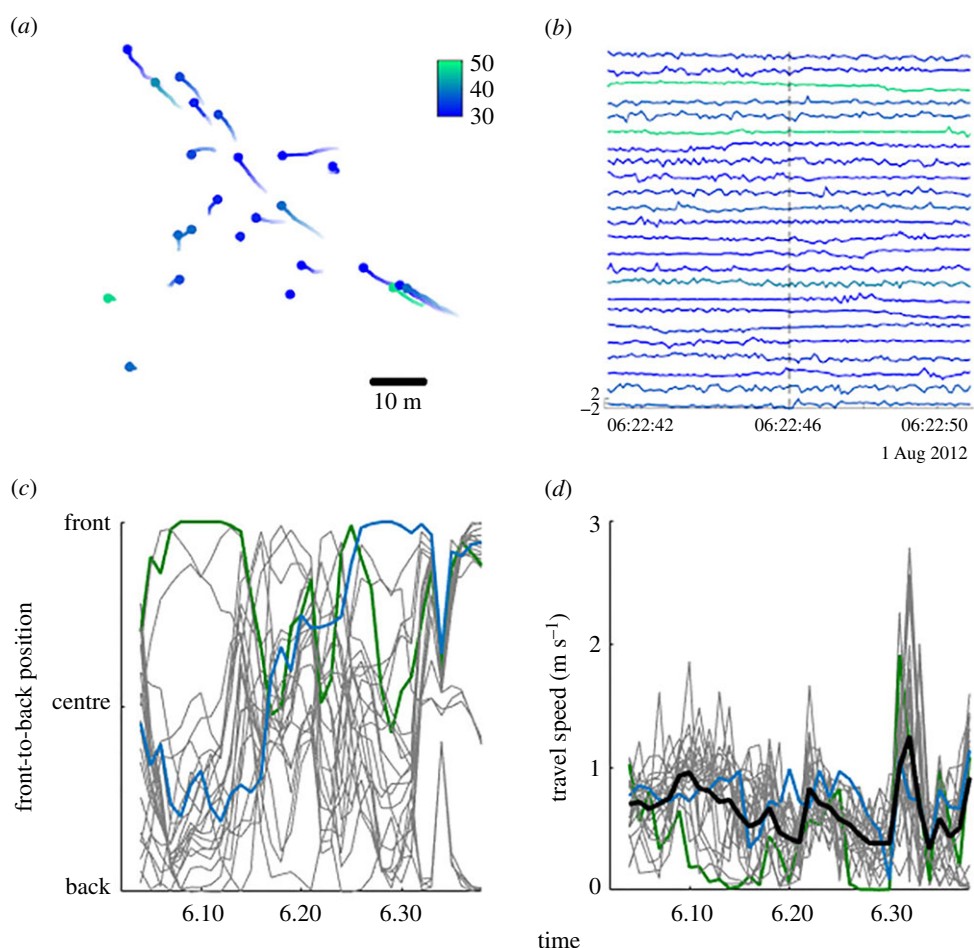

**Figure 1.** Characterizing the movement of a group of baboons. A snapshot of (*a*) the locations of baboons at time *t*, represented by circles, with tails stretching back to individuals' locations at *t*-10 s, and (*b*) the heave-axis acceleration—with peaks representing footfalls—of all individuals, show variation in baboons' move and pause activity states, as well as in their stride frequencies. Over a 40-min period, (*c*) individuals' position within the group, relative to the direction of group movement, as well as (*d*) individuals' speeds during group travel, are highly variable. In (*a*) and (*b*), line colour indicates leg length (in centimetres). The coloured lines on (*c*) and (*d*) highlight the patterns of two individuals. The thick black line in (*d*) represents the travel speed of the group centroid.

accumulation of GPS positional error inflating our estimates, we calculated daily travel distances after discretizing the data to 5-m resolution [32]. Daily maximum displacement from the sleeping site was measured as a straight-line distance between the group's morning sleeping site and the most distant position visited on that day. We used linear mixed models (LMMs) to estimate the effects of leg length on (i) daily travel distance and (ii) daily displacement, considering individual identity as a random effect and temporal autocorrelation between days using an autoregressive (AR1) component in both of the models [33].

The vectorial dynamic body acceleration (VeDBA) of each group member was calculated using data from tri-axial accelerometers, following Halsey *et al.* [34] and Wilson *et al.* [35]. Derivatives of dynamic body acceleration, such as VeDBA and ODBA (overall dynamic body acceleration), are proxy measures for movement-based energetic expenditure that has been validated for several quadrupedal taxa [34–37]. Because we did not perform calibration experiments, we did not aim to use VeDBA to quantify the actual energy expenditure [38] but instead used it as a proxy to compare between the different group activity states and among individuals.

## (d) Identifying local decision rules and their influence on cohesion

To assess how the local decision rules that individuals make with respect to modulating their travel speed change the collective

properties of their group, we compared model simulations to our observed data. We modelled group spread under three alternative scenarios for a group moving in one dimension. The scenarios included individuals that (i) moved at their characteristic speed, (ii) changed their speed as function of their position in the group or (iii) moved at their characteristic speed when group spread was low and changed their speed as function of their position when the spread exceeded a threshold value. The durations of simulations were drawn from the distribution of the observed group travel bouts. Models were ranked according to their AIC scores [39]. To estimate the emergent spatial segregation, we sampled the relative location of large and small individuals and calculated the front-to-back positional rank difference between the two size categories. In the empirical data, individuals' relative positional ranks on the front-to-back axis were determined by multiplying their *x*–*y* locations by a rotation matrix based on the heading of the centroid. We used an LMM to test the effect of body size on the positional rank difference with the group movement event as a random effect, for both simulated and observed movement tracks. To obtain a reliable estimation of the front-to-back rank position of group members we limited this and the following analyses to group travel bouts when the group centroid was moving for at least two minutes and data were available for at least 16 baboons (variation caused by collar dropout; see [40] for details).

We assessed how a focal individual's leg length, its position within the group (as a linear and quadratic term), the difference in leg length between the focal individual and its nearest

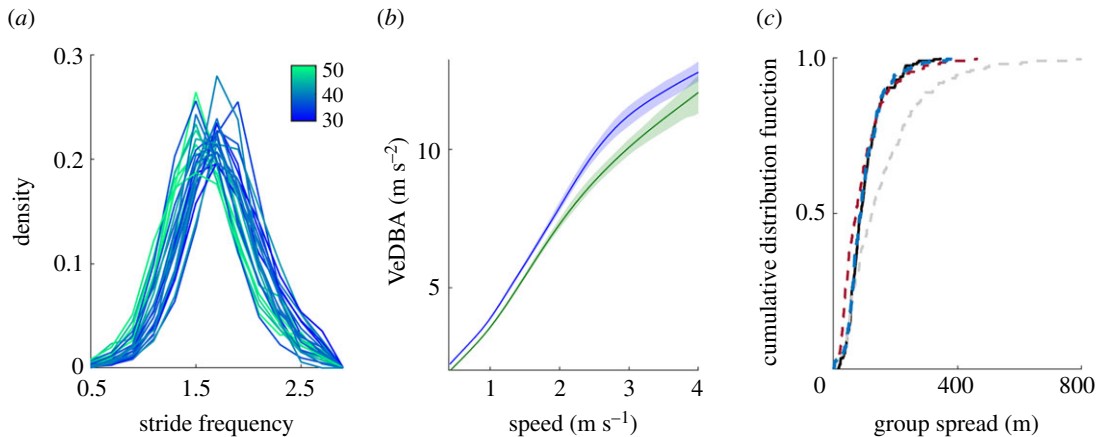

**Figure 2.** Body size affects locomotion. (*a*) Variation in individuals' stride frequency during independent movement. Line colour indicates leg length (in centimetres). (*b*) Differences in movement costs as estimated by VeDBA. Smaller individuals had higher VeDBA compared to large individuals travelling at the same speed. Individuals with longer leg length than average are represented in green, and shorter than average, in blue. (*c*) Group spread is affected by speed modulation decisions. Cumulative distribution function for group spread under four alternative scenarios: observed (black solid), individuals moving at their characteristic speed (grey dashed), characteristic speed modulated by position (red dashed), and characteristic speed modulated by position and spread (blue dashed).

**Table 1.** A summary list of derived variables with short explanations.

| predictor | description |
| --- | --- |
| characteristic stride frequency | an individual's average stride frequency while moving alone, i.e. a reference value that represents the stride frequency which she/he chooses, independent of the need to maintain cohesion |
| stride frequency deviance | the deviation in stride frequency from the characteristic stride frequency at any given time when the group was non-stationary |
| move : pause ratio | the ratio of time spent moving to time spent stationary when the group was non-stationary |
| VeDBA | a proxy measure derived from acceleration data for movement-based energetic expenditure, calculated when the group was non-stationary |
| daily travel distance | an estimate of the total distance travelled by an individual during a day. Location data were spatially discretized at 5 m intervals to reduce the impact of measurement error |
| daily maximum displacement | straight-line distance between the group's morning sleeping site and the most distant position visited on that day |

neighbour (for a subset of cases in which the nearest neighbour distance was under 5-m), group speed, and the group spread affected focal individuals' (i) stride frequency deviation and (ii) VeDBA (table 1 for a summary of the response variables). For this analysis, the front-to-back positions were rescaled such that—regardless of the group spread—1 represents being at the front, 0 represents the centre, and −1 represents the back of the group (figure 1*c*). To account for the dynamic nature of both predictor and response variables within a travel bout, all measures were aggregated over 10 s intervals. The candidate LMMs included all of the above factors as main effects and also an interaction term between leg length and position in the group. All models accounted for individual identity and the cohesive group movement event as crossed random factors, and considered temporal autocorrelation by using an autoregressive moving average (ARMA) component. We also examined how an individual's position in the group affected its activity states. We calculated the ratio of time spent moving to time spent stationary, henceforth the move : pause ratio. We used generalized linear mixed models (GLMMs) with beta error distribution and logit link function and the predictors, focal individual's leg length and its front-to-back position as fixed effects. These models also accounted for

individual identity and the group movement event as crossed random factors. Data analyses were performed in Matlab and R [41] using the packages nlme and MuMin [42,43].

## 3. Results

### (a) Locomotor capacity varies with body size

Baboons in the study group varied substantially in size (leg length: mean = 38 cm, range = 31–51 cm), stride frequencies and movement patterns. Individuals displayed characteristic stride frequencies that varied as a function of body size (figure 2*a*): when the group was stationary and individuals moved independently about the stationary group, stride frequency was negatively correlated with leg length (Pearson correlation, $r = -0.53$, $p = 0.01$). Variation in stride frequencies extended to the context of collective movement. When the group was moving cohesively ($N = 96$ travel bouts lasting $26 \pm 2$ min; mean + s.e.), larger individuals exhibited lower

stride frequencies than smaller group members (LMM, $b \pm$ s.e. $= 0.013 \pm 0.002$ Hz, Wald $t = 2.1$, $p = 0.015$).

Baboons of different sizes also varied in their movement patterns; high-resolution GPS tracking revealed significant inter-individual variation in total daily distances travelled. The group as a whole—measured from the position of its centroid—travelled a mean of 7.2 ($\pm 1.8$ s.d.) km per day, with most of that distance covered during long travel bouts, punctuated by periods when the group remained relatively stationary. Individual baboons travelled for $142 \pm 25$ min each day, during which they covered $8.4 \pm 1.2$ (mean $\pm$ s.d.) km. Individual daily travel distance was negatively related to body size (Wald $t = 2.3$, $p = 0.008$), and decreased 30 ($\pm 10$ s.d.) m with each 1 cm increase in leg length. Only minor differences ($\pm 1\%$) were found among individuals' daily maximum displacement from the group sleeping site, reflecting their shared travel route.

Differences in individual locomotion and movement patterns had consequences that disproportionately impacted smaller individuals, particularly when the group engaged in collective movement. Overall, VeDBA decreased with increasing body size ($b \pm$ s.e. $= -0.15 \pm 0.04$ m s$^{-2}$ for each 1 cm change in leg length; Wald $t = 3.93$, $p < 0.001$) and increased with travel speed ($b \pm$ s.e. $= 6.15 \pm 0.01$, Wald $t = 76.6$, $p < 0.001$, figure 2b). Increases in travel speed had a larger impact on VeDBA of smaller baboons compared to their larger group-mates ($b \pm$ s.e. $= 0.14 \pm 0.03$, Wald $t = 3.89$, $p < 0.001$). VeDBA values were higher when individuals moved together compared to when they were moving and the group was stationary ($b \pm$ s.e. $= 0.05 \pm 0.005$, Wald $t = 9.74$, $p < 0.001$).

## (b) Socially mediated movement decisions maintain group cohesion

We compared observed patterns of the group spread and size-based spatial segregation to patterns predicted by simulations where individuals varied their stride frequency as a function of their leg length. Simulations in which individuals moved without any modulation to their characteristic stride frequency overestimated group spread by eight-fold [$\Delta$AIC $= 260$]. By contrast, the incorporation of simple socially based decision rules improved model performance. The addition of a single rule in which individuals vary their speed as a function of their position within the group provided a good fit to our observed data for long travel bouts, but underestimated group spread for short travel bouts by two-fold [$\Delta$AIC $= 220$]. The best-fitting model incorporated position-dependent modulation of speed when group spread was larger than a threshold value (estimated to be 80 m), but allowed individuals to move at their characteristic stride frequency when the group was highly cohesive (figure 2c). These patterns align with our empirical data, where the mean deviation of group members from their characteristic stride frequency increased by a mean of 0.7% ($\pm 0.2\%$ s.d.) with 1 m increase in group spread (Wald $t = 4.51$, $p < 0.001$; figure 3e). Our simulations predict that size-based segregation will emerge if group members move at their characteristic stride frequency and do not modulate their stride frequency based on their position in the group; in simulated travel bouts, front-to-back positional rank was positively associated with body size (LM; $b \pm$ s.e. $= 2.71 \pm 0.90$, Wald $t = 2.47$, $p = 0.005$). By contrast,

we found no evidence for size-based segregation in our empirical travel bouts; the mean rank difference between large and small individuals ($b \pm$ s.e. $= 0.46 \pm 0.85$) was not distinguishable from zero.

## (c) Local decision rules support the emergence of cohesion

Baboons modulate their travel speed by varying their stride frequency and their move : pause ratios. An individual's decision to adjust these fine-scale movement characteristics was sensitive to its social context. Individuals adjusted their stride frequency depending on the relative size of their nearest neighbour. Relative to their characteristic stride frequency, baboons increased their stride frequency when travelling in proximity (less than 5 m) to larger individuals, and decreased their stride frequency when in proximity to smaller individuals. However, the size of these behavioural adjustments was not equal; smaller individuals increased their stride frequency more than their larger neighbours decreased their stride frequency (LMM; $b \pm$ s.e. $= 0.13 \pm 0.04\%$, Wald $t = 2.68$, $p = 0.003$; figure 3b).

Position within the group also influenced baboons' movement decisions. While individuals in the front of the group maintained their characteristic stride frequency, baboons, regardless of size, increased their stride frequency when they were at the back ($b \pm$ s.e. $= 1.6 \pm 0.3\%$, Wald $t = 2.70$, $p = 0.002$). However, the behavioural strategies of small and large individuals differed at the centre of the group (figure 3b). In these central positions, smaller individuals increased their stride frequencies but larger individuals did not deviate from their characteristic stride frequency (interaction between the quadratic term of group position and leg length; $2 \pm 0.2\%$, Wald $t = 2.14$, $p = 0.015$). Baboons changed their position within the group regularly (figure 1c), maintaining the same positional rank along the front-back axis for an average of only $54.0 \pm 14.1$ s. Overall, smaller individuals exhibited higher move : pause ratios than their larger group members (binomial GLMM, $b \pm$ s.e. $= 0.46 \pm 0.10$, Wald $z = 4.41$, $p < 0.001$), but this was especially true when they were at the back of the group (figure 3a). All group members were less likely to move when they were at the front of the group, and more likely to move when they were at the back. The spatial scale at which separation from the group prompted a change in an individual's move : pause ratio differed depending on whether an animal had outstripped or had fallen behind the rest of the group. At the front of the group, the increase in move : pause ratios occurred when individuals got 20 m ahead of their group-mates, whereas individuals had to fall at least 40 m behind the rest of their group before increasing their move : pause ratios (figure 3d).

## 4. Discussion

In social species, variation in individual locomotor capacity complicates collective movement by forcing group members to modulate their speed in order to maintain group cohesion. Our simulations demonstrate that to replicate the levels of cohesion we observe in wild animal groups, group members need to dynamically adjust their patterns of movement in response to their social context. Simultaneous tracking of the majority of a group of wild baboons using GPS and

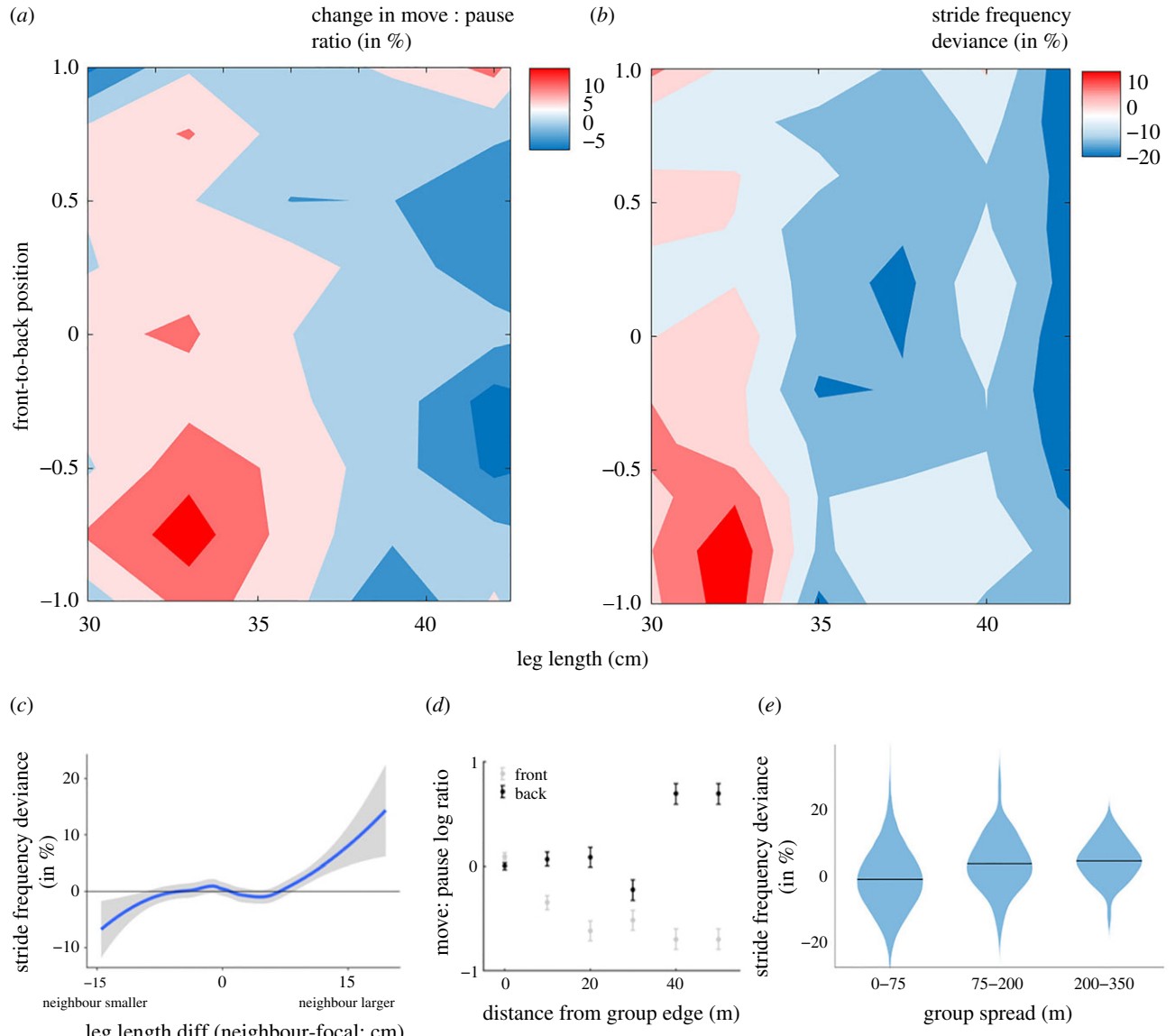

**Figure 3.** Context-dependent decision rules support the emergence of cohesion. (*a*) The move : pause ratio and (*b*) stride frequency deviation varied depending on an individual's leg length and front-to-back position in the group. (*c*) Individuals modulate their locomotor behaviour based on size-asymmetries with their nearest neighbours. Focal individuals increased their stride frequency when travelling near larger group-mates, and decreased their stride frequency when travelling near smaller group-mates, albeit to a lesser degree. (*d*) Patterns of intermittent locomotion varied as a function of position within the group, but the effect was asymmetrical. Individuals changed their move : pause ratios when separated from their group, but were more sensitive to being out in front than to falling behind. (*e*) Stride frequency deviation was lower when group spread was small (0–75 m) compared to larger group spread (75–150 m, 150–350 m; electronic supplementary material, figure S1).

accelerometer data loggers provided an opportunity to assess how individuals modulate their fine-scale behaviour in response to changes in their social environment, and thereby maintain the spatial cohesion of their group. Individuals have a characteristic stride frequency that relates to their body size, but they adjust this stride to match the pace of movement of their nearest neighbours. Furthermore, individuals deviate more from their characteristic stride frequency when group spread increases. Individuals also balanced their tendency to pause during group movement as a function of their spatial position within the group, waiting when they outstripped the group, and hustling to catch up when they fell behind.

While all group members modulated their movement patterns in these ways, they did so to differing degrees. Compared to other members of their group, small baboons showed larger deviations from their characteristic stride

frequency (figure 3*b*). Consistent with previous work suggesting that changes in gait characteristics have important energetic consequences [5], smaller baboons also spent more time moving and had higher VeDBA (i.e. a proxy for energetic expenditure) than their larger group-mates. Size-based differences in VeDBA were magnified as travel speed increased (figure 2*b*). Stride frequency for a given speed increases with decreasing animal size, in part due to decreasing leg length [6], which has also been invoked to explain lower mass-specific costs of transport in larger animals [44]. Our observations of increased changes in stride frequency in smaller animals indicate a similar pattern and are also, unsurprisingly, coupled with higher VeDBA values. While size-related differences in VeDBA may be partially driven by differences in body mass or collar attachment, our results suggest that small individuals pay disproportionate costs associated with maintaining group cohesion. Small

individuals are expected to incur additional costs if the effort required to keep up with their group-mates decreases their foraging efficiency—a likely outcome if small individuals are unable to pause to 'forage on the go'. Small group members may disproportionately bear the costs of cohesion because they have the most to gain from group membership. Indeed, smaller individuals are typically more vulnerable to predation in baboons [23] and other primates [45]. Moreover, small individuals are typically younger and less experienced [24], and therefore may stand to benefit more from shared information than their larger, older, more experienced counterparts. Our results are consistent with those from studies of shoaling fish, which show that fish from high-predation habitats maintain larger and more cohesive groups, with individuals modulating their acceleration and deceleration dynamics to maintain proximity to neighbours more in comparison with those from low-predation habitats [46,47]. This suggests that, as with the baboons in our study, animals who can gain the most protection against predators from group-living work the hardest to maintain group cohesion. Further work to test this hypothesis, explicitly measuring energetic costs and group-membership benefits across group members, is warranted.

Baboons modulate their fine-scale movement decisions differently depending on their spatial position within their troop. We showed that individuals at the front of the group are more sensitive to group spread and pause often to let the rest of their group catch up, while individuals at the back allow more separation from the group before increasing their move : pause ratios to catch up. This probably reflects context-dependent costs and benefits of different relative positions within the group [48]. Differences in spatial positioning create variation in the ability of group members to exert influence on group members [40,49,50] with the contribution of individuals at the back of the group compromised. However, the willingness of individuals at the back of the group to allow a larger separation from other group members may reflect the benefits incurred by pausing often for small foraging bouts (and thus falling behind), and may be enabled by a perceived lower risk of predation associated with this position. Conversely, when individuals outstrip their group, their decision to slow down likely reflects a trade-off between the opportunity costs of delaying arrival at their destination and the benefits of pausing to forage, as well as maintaining proximity to group-mates while in this particularly risky position within the group [51,52] Social dynamics may also influence individuals' decisions about the tempo of travel and how frequently to pause, and warrant additional attention in future research. For example, when male baboons are mate-guarding, the behaviour of both the estrus female and competing males are likely to influence movement decisions. Likewise, the movement decisions of mothers may be highly constrained by small, but independently locomoting offspring [53].

Individual group members showed significant variation in their daily travel distances. Because all members of the troop followed the same general route, these individual differences in travel distance result from variation in individuals' local, small-scale movements. In general, smaller individuals had longer daily travel distances, suggesting that body size may play a role in the tortuosity of the movement track. Inter-group and inter-population differences in baboon troop daily travel distances are well studied and can be attributed to a range of social and environmental factors including group size, food availability and local interactions with other baboon troops and other species [54,55]. However, the study of fine-scale variation in the daily travel distances of individuals in heterogeneous, socially cohesive groups is lacking. To our knowledge, there is no theoretical framework that explains why such differences arise. A more in-depth study of the effects of group members' body size, life-history traits and social context on the fine-scale differences in movement patterns is needed to understand why some individuals travel farther, even along the same route.

The differences in locomotor patterns documented in this study suggest that the consequences of collective movement vary among members of heterogeneous groups. It is well established that the ecological cost of transport is highly variable across species, ranging from 0.19% to 28% of overall energy expenditure [56], and that smaller animal species have higher energetic costs associated with locomotion [44]. While less is known about intraspecific relationships between body size and the energetic costs of locomotion, studies suggest that the same holds within species [11,57,58]. This is consistent with our results showing that VeDBA was higher for smaller individuals and that the relationship between increasing speed and increasing VeDBA scaled with body size, with smaller individuals having relatively higher increases in VeDBA over increasing speeds. Variation in energy expenditure can result not only from variation in travel speed, but also from differences in individuals' tendencies to move and pause [59]. However, energy expenditure encompasses only a part of an individual's energy balance. It is yet to be revealed how metabolic rates, energy intake, endurance and recovery dynamics change with body size [60,61], but all could potentially impact the cost of collective motion in heterogeneous groups. Tri-axial accelerometry provides a promising new method of quantifying many such inter-individual differences and may afford new opportunities for studying the costs of sociality in wild animals.

Given our findings, we expect that heterogeneity in movement costs among group members may affect group size and composition, especially in species that exhibit stable social groups. When small and large individuals have significant disparities in locomotor capacity such that smaller individuals simply cannot keep up with, or travel as far as, larger individuals over the course of full days, this could cause a reduction in groups' spatial cohesion, changes in fission–fusion social dynamics (e.g. [62]) and impose constraints on daily travel distances [63]. Individuals living in large groups are expected to obtain more information about the environment [13], have more reliable estimations of it [64,65], and experience reduced risk [15]. However, larger groups must travel farther when foraging [66,67], and longer daily travel distances exacerbate inequalities in the energetic costs of locomotion. Therefore, we predict that larger group size and increased heterogeneity in body size will lead to greater disparities in costs associated with collective movement, and stress the need of covering a broader range of group compositions and sizes in future studies.

The compromises that individuals make to maintain group cohesion occur across many axes—including compromises related to dietary, safety and social needs [10,18,68]. A holistic view that considers the interactions between these axes of compromise is necessary to understand how individuals

balance the costs and benefits of group-living. In olive baboons, group movement trajectories are steered by a process of shared decision-making among group members, suggesting that individuals may often make compromises in the timing and direction of movement in order to stay with their group [25]. In this study, we show that individuals modulate their fine-scale locomotor behaviours relative to their social context and spatial position within the group during collective movement. All group members thus make locomotor compromises to maintain group cohesion, suggesting that the *pace* of collective movement is also driven by a shared decision-making process. Our findings stress the importance of considering the interaction between social dynamics and locomotor capacity in shaping the movement ecology of group-living species, and illustrate an approach for accomplishing this under socially and ecologically relevant field conditions.

Ethics. All procedures were subject to ethical review and were carried out in accordance with the approved guidelines set out by the National Commission for Science, Technology and Innovation of the Republic of Kenya (NACOSTI/P/15/5727/4608) and with permission of the Kenya Wildlife Service (KWS/BRM/5001). Baboon tracking was approved by the Smithsonian Tropical Research Institute (IACUC 2012.0601.2015).

Data accessibility. Location and acceleration data used in the manuscript are available from Movebank Data Repository: https://doi.org/10.5441/001/1.3q2131q5 [69].

Authors' contributions. R.H.: Conceptualization, formal analysis, investigation, methodology, visualization, writing—original draft,

writing—review and editing; J.C.L.: Methodology, writing—review and editing; M.C.C.: Conceptualization, data collection, data curation, funding acquisition, investigation, project administration, supervision, writing—review and editing. All authors gave final approval for publication and agreed to be held accountable for the work performed therein.

Competing interests. We declare we have no competing interests.

Funding. We acknowledge funding from the Max Planck Institute for Animal Behavior, the Smithsonian Tropical Research Institute and University of California, Davis. R.H. and M.C.C. were supported by the National Science Foundation (grant nos. IIS 1514174 and IOS 1250895). M.C.C. received additional support from a Packard Foundation Fellowship (2016-65130), and the Alexander von Humboldt Foundation in the framework of the Alexander von Humboldt Professorship endowed by the Federal Ministry of Education and Research awarded to M.C.C. J.C.L. was supported by an NSF Graduate Research Fellowship and a UC Davis Dean's Distinguished Graduate Fellowship. Support was also provided by the Center for the Advanced Study of Collective Behavior at the University of Konstanz, DFG Centre of Excellence 2117 (ID: 422037984).

Acknowledgements. We thank Kenya National Science and Technology Council, Kenyan Wildlife Service and Mpala Research Centre for permission to conduct research. We are grateful to Alison Ashbury, Tanya Berger-Wolf, Damien Farine, Ariana Strandburg-Peshkin, Rory Wilson, Mark Grote, Yuuki Watanabe and four anonymous reviewers for helpful comments and suggestions. We thank Martin Wikelski, Eldredge Bermingham, Dan Rubenstein, Margaret Kinnaird, Diane Carlino, Roland Kays, Suzan Murray, Mathew Mutinda, Robert Lessnau, Shauhin Alavi, Julius Nairobi, Franz Kuemmeth and Wolfgang Heidrich for logistical and technological support, assistance with animal capture and field-based data collection.

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
