## [Peer Review File · Proceedings of the Royal Society B: Biological Sciences]

Review History

RSPB-2021-0839.R0 (Original submission)

Review form: Reviewer 1

Recommendation

Major revision is needed (please make suggestions in comments)

Scientific importance: Is the manuscript an original and important contribution to its field?

Good

General interest: Is the paper of sufficient general interest?

Acceptable

Quality of the paper: Is the overall quality of the paper suitable?

Good

Is the length of the paper justified?

Yes

Should the paper be seen by a specialist statistical reviewer?

No

Do you have any concerns about statistical analyses in this paper? If so, please specify them explicitly in your report.

No

It is a condition of publication that authors make their supporting data, code and materials available - either as supplementary material or hosted in an external repository. Please rate, if applicable, the supporting data on the following criteria.

Is it accessible?

No

Is it clear?

N/A

Is it adequate?

N/A

Do you have any ethical concerns with this paper?

No

Comments to the Author

This is an interesting article documenting the relatively understudied role of body size and location within a troop, in collective and individual movement patterns. The manuscript is well written and easy to follow, although I suggest below that the Methods section could be restructured slightly to make it clearer. I have also added some general comments about the rest of the article below.

The authors investigated the role of body size (leg length) on locomotion, however I wonder whether finer metrics of size should also be investigated, such as leg:body length ratios. Presumably the proportional size of limbs (relative to body size) would also matter in addition to just overall limb length. Can the authors please comment as to whether they expect results to vary if such metrics were investigated as well?

In the intro e.g. lines 60-2, the authors' main hypothesis for the need to coordinate movements within a group is in response to risks of predation. However, this is just one potential hypothesis for coordinated movements, another being to enhance foraging efficiency, the formation of sub-groups/coalitions, potential roles of kinship or parentage etc, in increasing benefits of coordination. Can the authors please discuss these alternative hypotheses or is there already very clear evidence that this is solely driven by predation (note that predation risk was not actually measured in this study).

Line 64-66 should be moved to the Methods section

Methods

By looking at just 1 group (which I understand the limitation), it does however mean that factors such as group size can not be accounted for in explaining differences in locomotory patterns between small and large. For eg, in smaller troops, predation risk may be higher hence the benefits to all individuals of altering gait etc would be greater, than in larger groups. This also needs to be mentioned in the discussion i.e. that a greater range of group sizes should be the aim of future studies.

The authors sampled 25 individuals, but do not mention what proportion of the group this was. The majority? What was overall troop size? And importantly, how many small versus large individuals were sampled? Was replication in this respect enough?

From then on the methods become quite difficult to follow owing to the number of response variables that were being analysed e.g. daily travel dist, daily displacement, stationary versus non, group spread, moving v non, heave peak frequency/amplitude, max power, stride frequency, VeDBA etc. Given the number of metrics involved, and the limited description/definition of each, I suggest the authors include a table which lists each of them and aside it a clear description of what they are. This will particularly help the non-specialist to understand the analyses more clearly as well.

Further, the methods jump between descriptions of group- versus individual-level metrics, and in some cases e.g. line 116 'individuals relative positions' seems more like an individual-level rather than group-level metric. I suggest re-structuring the methods so that all group and all individual level metrics are lumped together (or alternatively, use the same subheadings as per the Results section in the methods, and describe all relevant analyses under each).

Discussion

The effect of nearest neighbour speed is interesting, can authors elaborate on the social structure and whether say offspring/kin are more likely to be closer to parents/kin etc within the group? Also was there any evidence of coalitions within groups forming and travelling closer together?

Re the use of loggers - how heavy are the loggers and are they likely to disproportionately impede movement of smaller individuals, hence contributing to some of these size-differences?

Discussion needs to be broadened to encompass studies not just of baboons but other taxa. E.g. flocks birds during flight, schools of fish are 2 key examples of collective movement that come to mind. How do these results translate across a broader taxonomic scale? In fish, schooling tends to occur between similar sized individuals/phenotypically similar individuals, maybe part of the reason is so that they don't have to adjust their locomotion to accommodate smaller fish. Like baboons, fish form coordinated groups (ie. schools) for predation defence and sometimes for group foraging. So why don't baboons also form size homogeneous groups? what keeps these heterogeneous groups together? The answer may be as simple as the fact that they benefit from living with kin, but this needs to be discussed given the importance of locomotion modulation wrt to size in a range of taxa beyond mammals (where relatedness may play a limited role).

Review form: Reviewer 2

Recommendation

Accept with minor revision (please list in comments)

Scientific importance: Is the manuscript an original and important contribution to its field?

Excellent

General interest: Is the paper of sufficient general interest?

Excellent

Quality of the paper: Is the overall quality of the paper suitable?

Excellent

Is the length of the paper justified?

Yes

Should the paper be seen by a specialist statistical reviewer?

No

Do you have any concerns about statistical analyses in this paper? If so, please specify them explicitly in your report.

No

It is a condition of publication that authors make their supporting data, code and materials available - either as supplementary material or hosted in an external repository. Please rate, if applicable, the supporting data on the following criteria.

Is it accessible?

Yes

Is it clear?

Yes

Is it adequate?

Yes

Do you have any ethical concerns with this paper?

No

Comments to the Author

Here the authors examine the effects of inter-individual differences in body size on both individual and group movement in a social species. This is a fascinating topic and the manuscript present a rich combination of results that adds much, in my opinion, to the literature currently available on collective behavior. More specifically, the authors contribute novel findings on the locomotor compromises made by group members to maintain cohesion. This subject is thus of relevance to researchers interested in collective behavior as well as the origins and maintenance of sociality.

Beyond the appeal of the research question itself, the authors present a well-written manuscript that makes excellent use of pre-existing, publicly available data. I believe the one notable weakness of their dataset is the short time period of data collection. It is my understanding, however, that this short window is a direct consequence of the intensity of GPS and accelerometer data which they collected. Relevant to this point, I do think it would be useful to clarify how many tracking days were analyzed. I noted the reference to the start and end of the overall tracking window, but it is not clear how many days within this roughly 30-day window were included in their analyses.

Regarding suggestions for change and potential improvement, I have a few minor points to raise. First, it is unclear to me whether variation in gait, collar fit, etc. between individuals might complicate the approach used to distinguish moving and non-moving activity states (line 122). Presumably, the authors pooled video recording of a subset of collared individuals for this analysis and did not – at this stage – make comparisons between individuals. If so, is there any concern that differences in walking style between individuals and/or subtle differences in the fit of the collar around the baboon's neck might affect the results? If this is a relevant concern, I believe this might be largely corrected when examining the heave-axis peaks for each collared individual with the moving windows. Nonetheless, some clarification would be helpful.

Second, since size is influenced by both sex and age, it would be helpful to provide more details on how these two variables together affect risk vulnerability. For example, do the authors speculate that males and females of equal size (but presumably different ages) experience the same predation risk? Would age (as a proxy for experience) perhaps affect this to some degree? On a related note, providing results on how body size varied by sex and age in the study troop would be useful. My assumption is that sexes and ages had overlapping measures of limb length, but it would be valuable to state this explicitly.

Third, the authors acknowledge that life-history traits and social context might influence daily travel distances (line 377), but why would these not also be important considerations for move-pause dynamics, costs of cohesion, etc.? In other words, why is this not a point discussed with regard to more of the study's findings? As one consideration for discussion (not analysis, as I believe it would be beyond the scope of this study), I imagine that female reproductive state could have a strong effect on move-pause dynamics, particularly considering when dependent offspring were carried vs. traveled independently.

Fourth, I am curious about variation in gait length. The authors' analysis of stride frequency are exciting and, I believe, expertly executed. As one additional consideration, though, I wonder if the baboons might be altering gait length as well as stride frequency. Could this be tested using the collars' high resolution GPS data to look at displacement distance per x number of strides? Or would this be over-reaching the spatial accuracy of the GPS readings?

Finally, as a minor comment, the authors note that their results are "consistent with the hypothesis that the costs of maintaining cohesion are largely borne by individuals that have the most to gain from group membership" (lines 347-349). My understanding is that they are referencing the fact that certain individuals are disproportionately vulnerable to predation if left alone. Broadly speaking, the benefits of group membership are of course not exclusive to minimized risk of predation (e.g. enhanced reproductive opportunities). Could the authors thus please clarify exactly what they mean by "most to gain" in this statement?

Decision letter (RSPB-2021-0839.R0)

01-Jun-2021

Dear Dr Harel:

Your manuscript has now been peer reviewed and the reviews have been assessed by an Associate Editor. The reviewers' comments (not including confidential comments to the Editor) and the comments from the Associate Editor are included at the end of this email for your reference. As you will see, the reviewers and the Editors have raised some concerns with your manuscript and we would like to invite you to revise your manuscript to address them.

Research ethics:

Use of animals and field studies:

It is a condition of publication that you make available the data and research materials supporting the results in the article. Please see our Data Sharing Policies (<https://royalsociety.org/journals/authors/author-guidelines/#data>). Datasets should be deposited in an appropriate publicly available repository and details of the associated accession number, link or DOI to the datasets must be included in the Data Accessibility section of the article (<https://royalsociety.org/journals/ethics-policies/data-sharing-mining/>). Reference(s) to datasets should also be included in the reference list of the article with DOIs (where available).

Please submit a copy of your revised paper within three weeks. If we do not hear from you within this time your manuscript will be rejected. If you are unable to meet this deadline please let us know as soon as possible, as we may be able to grant a short extension.

Best wishes,
Dr Sasha Dall
mailto: proceedingsb@royalsociety.org

Associate Editor
Board Member: 1
Comments to Author:

The reviewers agreed that the manuscript is well written, addresses an important topic, and makes a substantial contribution to the literature. They had several suggestions for clarifying the methods section and broadening the discussion.

One issue came up that is very critical, but easy to address: data accessibility. One of the reviewers was not able to download the supplementary data, probably because it is a huge file (6GB). I was able to download it, but I could not open it fully because it is too big for my standard programs to handle (Excel, Notepad++), even on a high-end desktop. I suggest breaking this up into multiple smaller files - otherwise, it is not truly accessible without specialized computer programs/systems. Alternatively, if large file sizes are necessary and specialized software is required, please provide instructions in the supp. info. section.

The reviewers requested some clarifications to the methods, e.g., how many tracking days were analyzed, what proportion of the group was sampled, group level versus individual level metrics, etc. Please consider whether the methods and results sections could be restructured to make these things clearer.

Several points were brought up for discussion; both reviewers wondered how variables such as collar fit/weight and family groups might affect locomotion, and the role of group size and individual predation risk were of interest as well. Comparison with other animals that travel in groups, as suggested by Reviewer 1, could make the study relevant to a broader audience.

Reviewer(s)' Comments to Author:

Referee: 1

Comments to the Author(s)

This is an interesting article documenting the relatively understudied role of body size and location within a troop, in collective and individual movement patterns. The manuscript is well written and easy to follow, although I suggest below that the Methods section could be restructured slightly to make it clearer. I have also added some general comments about the rest of the article below.

The authors investigated the role of body size (leg length) on locomotion, however I wonder whether finer metrics of size should also be investigated, such as leg:body length ratios. Presumably the proportional size of limbs (relative to body size) would also matter in addition to just overall limb length. Can the authors please comment as to whether they expect results to vary if such metrics were investigated as well?

In the intro e.g. lines 60-2, the authors' main hypothesis for the need to coordinate movements within a group is in response to risks of predation. However, this is just one potential hypothesis for coordinated movements, another being to enhance foraging efficiency, the formation of sub-groups/coalitions, potential roles of kinship or parentage etc, in increasing benefits of coordination. Can the authors please discuss these alternative hypotheses or is there already very clear evidence that this is solely driven by predation (note that predation risk was not actually measured in this study).

Line 64-66 should be moved to the Methods section

Methods

By looking at just 1 group (which I understand the limitation), it does however mean that factors such as group size can not be accounted for in explaining differences in locomotory patterns between small and large. For eg, in smaller troops, predation risk may be higher hence the benefits to all individuals of altering gait etc would be greater, than in larger groups. This also needs to be mentioned in the discussion i.e. that a greater range of group sizes should be the aim of future studies.

The authors sampled 25 individuals, but do not mention what proportion of the group this was. The majority? What was overall troop size? And importantly, how many small versus large individuals were sampled? Was replication in this respect enough?

From then on the methods become quite difficult to follow owing to the number of response variables that were being analysed e.g. daily travel dist, daily displacement, stationary versus non, group spread, moving v non, heave peak frequency/amplitude, max power, stride frequency, VeDBA etc. Given the number of metrics involved, and the limited description/definition of each, I suggest the authors include a table which lists each of them and aside it a clear description of what they are. This will particularly help the non-specialist to understand the analyses more clearly as well.

Further, the methods jump between descriptions of group- versus individual-level metrics, and in some cases e.g. line 116 'individuals relative positions' seems more like an individual-level rather than group-level metric. I suggest re-structuring the methods so that all group and all individual level metrics are lumped together (or alternatively, use the same subheadings as per the Results section in the methods, and describe all relevant analyses under each).

Discussion

The effect of nearest neighbour speed is interesting, can authors elaborate on the social structure and whether say offspring/kin are more likely to be closer to parents/kin etc within the group? Also was there any evidence of coalitions within groups forming and travelling closer together?

Re the use of loggers - how heavy are the loggers and are they likely to disproportionately impede movement of smaller individuals, hence contributing to some of these size-differences?

Discussion needs to be broadened to encompass studies not just of baboons but other taxa. E.g. flocks birds during flight, schools of fish are 2 key examples of collective movement that come to mind. How do these results translate across a broader taxonomic scale? In fish, schooling tends to occur between similar sized individuals/phenotypically similar individuals, maybe part of the reason is so that they don't have to adjust their locomotion to accommodate smaller fish. Like baboons, fish form coordinated groups (ie. schools) for predation defence and sometimes for group foraging. So why don't baboons also form size homogeneous groups? what keeps these heterogeneous groups together? The answer may be as simple as the fact that they benefit from living with kin, but this needs to be discussed given the importance of locomotion modulation wrt to size in a range of taxa beyond mammals (where relatedness may play limited role).

Referee: 2

Comments to the Author(s)

Here the authors examine the effects of inter-individual differences in body size on both individual and group movement in a social species. This is a fascinating topic and the manuscript presents a rich combination of results that adds much, in my opinion, to the literature currently available on collective behavior. More specifically, the authors contribute novel findings on the locomotor compromises made by group members to maintain cohesion. This subject is thus of relevance to researchers interested in collective behavior as well as the origins and maintenance of sociality.

Beyond the appeal of the research question itself, the authors present a well-written manuscript that makes excellent use of pre-existing, publicly available data. I believe the one notable weakness of their dataset is the short time period of data collection. It is my understanding, however, that this short window is a direct consequence of the intensity of GPS and accelerometer data which they collected. Relevant to this point, I do think it would be useful to clarify how many tracking days were analyzed. I noted the reference to the start and end of the overall tracking window, but it is not clear how many days within this roughly 30-day window were included in their analyses.

Regarding suggestions for change and potential improvement, I have a few minor points to raise. First, it is unclear to me whether variation in gait, collar fit, etc. between individuals might complicate the approach used to distinguish moving and non-moving activity states (line 122). Presumably, the authors pooled video recording of a subset of collared individuals for this analysis and did not – at this stage – make comparisons between individuals. If so, is there any concern that differences in walking style between individuals and/or subtle differences in the fit of the collar around the baboon's neck might affect the results? If this is a relevant concern, I believe this might be largely corrected when examining the heave-axis peaks for each collared individual with the moving windows. Nonetheless, some clarification would be helpful.

Second, since size is influenced by both sex and age, it would be helpful to provide more details on how these two variables together affect risk vulnerability. For example, do the authors speculate that males and females of equal size (but presumably different ages) experience the same predation risk? Would age (as a proxy for experience) perhaps affect this to some degree? On a related note, providing results on how body size varied by sex and age in the study troop would be useful. My assumption is that sexes and ages had overlapping measures of limb length, but it would be valuable to state this explicitly.

Third, the authors acknowledge that life-history traits and social context might influence daily travel distances (line 377), but why would these not also be important considerations for move-pause dynamics, costs of cohesion, etc.? In other words, why is this not a point discussed with regard to more of the study's findings? As one consideration for discussion (not analysis, as I believe it would be beyond the scope of this study), I imagine that female reproductive state could have a strong effect on move-pause dynamics, particularly considering when dependent offspring were carried vs. traveled independently.

Fourth, I am curious about variation in gait length. The authors' analysis of stride frequency are exciting and, I believe, expertly executed. As one additional consideration, though, I wonder if the baboons might be altering gait length as well as stride frequency. Could this be tested using the collars' high resolution GPS data to look at displacement distance per x number of strides? Or would this be over-reaching the spatial accuracy of the GPS readings?

Finally, as a minor comment, the authors note that their results are "consistent with the hypothesis that the costs of maintaining cohesion are largely borne by individuals that have the most to gain from group membership" (lines 347-349). My understanding is that they are referencing the fact that certain individuals are disproportionately vulnerable to predation if left alone. Broadly speaking, the benefits of group membership are of course not exclusive to minimized risk of predation (e.g. enhanced reproductive opportunities). Could the authors thus please clarify exactly what they mean by "most to gain" in this statement?

Author's Response to Decision Letter for (RSPB-2021-0839.R0)

See Appendix A.

Decision letter (RSPB-2021-0839.R1)

02-Jul-2021

Dear Dr Harel

I am pleased to inform you that your manuscript entitled "Locomotor compromises maintain group cohesion in baboon troops on the move" has been accepted for publication in Proceedings B.

Data Accessibility section

Open Access

You are invited to opt for Open Access, making your freely available to all as soon as it is ready for publication under a CCBY licence. Our article processing charge for Open Access is £1700. Corresponding authors from member institutions (<http://royalsocietypublishing.org/site/librarians/allmembers.xhtml>) receive a 25% discount to these charges. For more information please visit <http://royalsocietypublishing.org/open-access>.

Paper charges

Sincerely,

Dr Sasha Dall

Appendix A

Dear Editor,

Many thanks for the constructive reviews and editorial feedback, which helped us to significantly improve our manuscript. Below we provide a detailed point-by-point response to the comments of all reviewers. We followed the helpful comments, clarifying the Methods and expanding the Discussion. We adopted most of the suggested changes of the reviewers and improved wording where reviewers pointed to confusing statements in the manuscript. We hope you will find our revision suitable for publication in *Proceedings B*.

Best wishes,

Roi Harel, Carter Loftus and Meg Crofoot

Associate Editor

Board Member: 1

Comments to Author:

The reviewers agreed that the manuscript is well written, addresses an important topic, and makes a substantial contribution to the literature. They had several suggestions for clarifying the methods section and broadening the discussion.

One issue came up that is very critical, but easy to address: data accessibility. One of the reviewers was not able to download the supplementary data, probably because it is a huge file (6GB). I was able to download it, but I could not open it fully because it is too big for my standard programs to handle (Excel, Notepad++), even on a high-end desktop. I suggest breaking this up into multiple smaller files - otherwise, it is not truly accessible without specialized computer programs/systems. Alternatively, if large file sizes are necessary and specialized software is required, please provide instructions in the supp. info.

Response 1. We followed your advice and added a Readme file on the repository accompanying the data files that specifies how the data should be downloaded and mentions the file size issue. The ReadMe file will provide instructions on how to download a small amount of data. We don't think that breaking the data to files would help because the files will still remain large (100s of MB). We recommend the use of a programming language, such as R, Python, or Matlab, to explore and analyze the data and the software FireTail for visualization purposes.

The reviewers requested some clarifications to the methods, e.g., how many tracking days were analyzed, what proportion of the group was sampled, group level versus individual level metrics, etc. Please consider whether the methods and results sections could be restructured to make these things clearer.

Response 2. We edited the structure of the Methods section, adding a table that presents the different aspects of behavior we analyze (Table 1), and a table presenting individual metadata (Table S1) following the helpful suggestions of both reviewers.

Collared individuals consisted of 54% (25/46) of group members (and 80% of the adults and sub-adults), including 13 adults, 10 sub-adults and two juveniles. Out of the 1 month of tracking, we used data from the first two weeks, resulting in 255 whole baboon days that were analyzed in the study (this information is now included at L103-107).

Several points were brought up for discussion; both reviewers wondered how variables such as collar fit/weight and family groups might affect locomotion, and the role of group size and individual predation risk were of interest as well. Comparison with other animals that travel in groups, as suggested by Reviewer 1, could make the study relevant to a broader audience.

Response 3. We find the feedback to be extremely helpful and made substantial changes in the main text.

To follow ethical guidelines, we used two sizes of tags in this study, keeping tag weight below the recommended cut-off of an individual's body weight. Relevant summary statistics of the relative weight can be found in the Methods (L128) and the relevant metadata regarding collar weight and body mass for the individuals can be found with the data on the Movebank repository. Collars were fitted in the same manner to all animals to minimize differences between individuals stemming from any impact of collars on movement patterns. When assessing stride frequencies, we examine the timings of the heave-axis peaks, rather than the amplitudes or other metrics which are likely to be more strongly affected by differences in gait and collar fit. Moreover, we observed the relative change in stride frequency from each individual's characteristic stride frequency and therefore inherently correct for differences between individuals.

We revisited the way we treat predation in the manuscript. We now note the benefits of grouping more broadly, including increased information, increased foraging efficiency and decreased predation risk (L60-66).

We acknowledge the need of the work to address different taxa that travel in groups to make it relevant to a broader audience. We rewrote several parts of the Discussion to compare our findings to works on other taxa. Specifically, we contextualized our study within the broader work on birds (Lees 2012, Papageorgiou 2020, Sankey 2019), fish (Couzin 2011, Herbert-Read 2017, Ioannou 2015, 2017; Jolles 2020, Mclean 2018), other mammals (Williams 2014) and other primates (Clutton-Brock 1977, Markham 2017, Petit 2009, Pontzer 2006, 2011).

Reviewer(s)' Comments to Author:

Referee: 1

Comments to the Author(s)

This is an interesting article documenting the relatively understudied role of body size and location within a troop, in collective and individual movement patterns. The manuscript is well written and easy to follow, although I suggest below that the Methods section could be restructured slightly to make it clearer. I have also added some general comments about the rest of the article below.

See Response 2.

The authors investigated the role of body size (leg length) on locomotion, however I wonder whether finer metrics of size should also be investigated, such as leg:body length ratios. Presumably the proportional size of limbs (relative to body size) would also matter in addition to just overall limb length. Can the authors please comment as to whether they expect results to vary if such metrics were investigated as well?

Response 4. We chose to use the metric of leg length based on previous work on biomechanics of terrestrial animals that found that the effective leg length is a primary anatomical driver of locomotor costs in terrestrial animals (Pontzer, 2007; Reilly et al., 2007). This contextualization was added to the main text (L101-103).

In the intro e.g. lines 60-2, the authors' main hypothesis for the need to coordinate movements within a group is in response to risks of predation. However, this is just one potential hypothesis for coordinated movements, another being to enhance foraging efficiency, the formation of sub-groups/coalitions, potential roles of kinship or parentage etc, in increasing benefits of coordination. Can the authors please discuss these alternative hypotheses or is there already very clear evidence that this is solely driven by predation (note that predation risk was not actually measured in this study).

See Response 3.

Line 64-66 should be moved to the Methods section

Response 5. We agree that this sentence was an overly detailed description of our methods for the introduction, and we have replaced it with a more general sentence about our observational approach (L67-68).

Methods

By looking at just 1 group (which I understand the limitation), it does however mean that factors such as group size can not be accounted for in explaining differences in locomotory patterns between small and large. For eg, in smaller troops, predation risk may be higher hence the benefits to all individuals of altering gait etc would be greater, than in larger groups. This also needs to be mentioned in the discussion i.e. that a greater range of group sizes should be the aim of future studies.

Response 6. Following the reviewer's suggestion, we rewrote the part in the Discussion that dealt with group size, and suggested the study of the effect of group size on our findings as an interesting direction for future work (L366-379).

The authors sampled 25 individuals, but do not mention what proportion of the group this was. The majority? What was overall troop size? And importantly, how many small versus large individuals were sampled? Was replication in this respect enough?

Response 7. We edited the Methods. See Response 2 regarding the details we added on collaring effort and group characteristics. Morphological measurement and age-sex class classification of all tracked group members can be found in Table S2 and the variation in body size is also visualized in Fig 1a and 2a. The sampling of different body sizes is not uniform, but represent a variety of different body sizes across the age and sex classes.

From then on the methods become quite difficult to follow owing to the number of response variables that were being analysed e.g. daily travel dist, daily displacement, stationary versus non, group spread, moving v non, heave peak frequency/amplitude, max power, stride frequency, VeDBA etc. Given the number of metrics involved, and the limited description/definition of each, I suggest the authors include a table which lists each of them and aside it a clear description of what they are. This will particularly help the non-specialist to understand the analyses more clearly as well.

Response 8. We liked the reviewer's suggestion and added a table (Table 1) which lists all the response variables with short definitions.

Further, the methods jump between descriptions of group- versus individual-level metrics, and in some cases e.g. line 116 ;individuals relative positions' seems more like an individual-level rather than group-level metric. I suggest re-structuring the methods so that all group and all individual level metrics are lumped together (or alternatively, use the same subheadings as per the Results section in the methods, and describe all relevant analyses under each).

Response 9. We followed the reviewer's suggestion and completely reorganized the methods to more closely match the structure of the Results section, with the appropriate subheadings matching those within the Results.

Discussion

The effect of nearest neighbour speed is interesting, can authors elaborate on the social structure and whether say offspring/kin are more likely to be closer to parents/kin etc within the group? Also was there any evidence of coalitions within groups forming and travelling closer together?

Response 10. We agree that the nearest neighbour effect the reviewer raises is extremely interesting and that social relations and kinship may have an important role in defining walking preferences. Unfortunately, data on kinship are unavailable during this particular data collection. We have recently returned to collect more data -- social/kin data as well as sensor data -- and the point that the reviewer mentions here is in the scope of another analysis we will be working on with the new data on coordination and synchronization between group members when the group is on the move.

Re the use of loggers - how heavy are the loggers and are they likely to disproportionately impede movement of smaller individuals, hence contributing to some of these size-differences?

See Response 3.

Discussion needs to be broadened to encompass studies not just of baboons but other taxa. E.g. flocks birds during flight, schools of fish are 2 key examples of collective movement that come to mind. How do these results translate across a broader taxonomic scale? In fish, schooling tends to occur between similar sized individuals/phenotypically similar individuals, maybe part of the reason is so that they don't have to adjust their locomotion to accommodate smaller fish.

See Response 3.

Like baboons, fish form coordinated groups (ie. schools) for predation defence and sometimes for group foraging. So why don't baboons also form size homogeneous groups? what keeps these heterogeneous groups together? The answer may be as simple as the fact that they benefit from living with kin, but this needs to be discussed given the importance of locomotion modulation wrt to size in a range of taxa beyond mammals (where relatedness may play limited role).

Response 12. We added a short explanation on why baboons live in groups, which contextualizes why these groups are inherently heterogeneous: “Females are philopatric and gain substantial fitness benefits from living with female kin and from the social bonds that they form with these kin (Silk et al. 2003). Residing in stable groups confers a range of benefits above and beyond the decreased predation risk, information exchange and foraging benefits that arise from aggregating with conspecifics (Krause & Ruxton 2002). In addition to dominance hierarchies and nepotistic kin relations, strongly differentiated social bonds structure baboon societies, and these ‘friendships’ provide significant fitness benefits for group members (Silk et al. 2009)”. (L70-77). See Response 3, regarding locomotion modulation in other taxa.

Referee: 2

Comments to the Author(s)

Here the authors examine the effects of inter-individual differences in body size on both individual and group movement in a social species. This is a fascinating topic and the manuscript presents a rich combination of results that adds much, in my opinion, to the literature currently available on collective behavior. More specifically, the authors contribute novel findings on the locomotor compromises made by group members to maintain cohesion. This subject is thus of relevance to researchers interested in collective behavior as well as the origins and maintenance of sociality.

Beyond the appeal of the research question itself, the authors present a well-written manuscript that makes excellent use of pre-existing, publicly available data. I believe the one notable weakness of their dataset is the short time period of data collection. It is my understanding, however, that this short window is a direct consequence of the intensity of GPS and accelerometer data which they collected. Relevant to this point, I do think it would be useful to clarify how many tracking days were analyzed. I noted the reference to the start and end of the overall tracking window, but it is not clear how many days within this roughly 30-day window were included in their analyses.

See Response 2.

Regarding suggestions for change and potential improvement, I have a few minor points to raise. First, it is unclear to me whether variation in gait, collar fit, etc. between individuals might complicate the approach used to distinguish moving and non-moving activity states (line 122). Presumably, the authors pooled video recording of a subset of collared individuals for this analysis and did not – at this stage – make comparisons between individuals. If so, is there any concern that differences in walking style between individuals and/or subtle differences in the fit of the collar around the baboon's neck might affect the results? If this is a relevant concern, I believe this might be largely corrected when examining

the heave-axis peaks for each collared individual with the moving windows. Nonetheless, some clarification would be helpful.

See Response 3. We moved this section to the supplementary material and added a clarification to the to address concerns that the classifier trained on a subset of individuals may not extend well to other individuals (L123-125).

Second, since size is influenced by both sex and age, it would be helpful to provide more details on how these two variables together affect risk vulnerability. For example, do the authors speculate that males and females of equal size (but presumably different ages) experience the same predation risk? Would age (as a proxy for experience) perhaps affect this to some degree? On a related note, providing results on how body size varied by sex and age in the study troop would be useful. My assumption is that sexes and ages had overlapping measures of limb length, but it would be valuable to state this explicitly.

Response 15. We find the ideas raised by the reviewer very interesting. Unfortunately, there is rather scarce data on predation in baboons, as predation events are difficult to observe due to the human-averse nature of their stealth predators. Thus, we have little information upon which to ground a hypothesis about how predation risk changes with age and sex. Upon further reflection, we decided that, due to the knowledge gap surrounding predation in baboons, we would remove some of the emphasis on predation within the manuscript adding the broader framing of the benefits of cohesion (Response 3). Information on age and sex variation in body size can be found in table 2S.

Third, the authors acknowledge that life-history traits and social context might influence daily travel distances (line 377), but why would these not also be important considerations for move-pause dynamics, costs of cohesion, etc.? In other words, why is this not a point discussed with regard to more of the study's findings? As one consideration for discussion (not analysis, as I believe it would be beyond the scope of this study), I imagine that female reproductive state could have a strong effect on move-pause dynamics, particularly considering when dependent offspring were carried vs. traveled independently.

Response 16. We agree with the comment and broaden the scope of our claim in the manuscript and state that we need to study variation in fine-scale movement decisions to understand better larger scale patterns, such as how far individuals travel. We added the point raised by the reviewer on the effect of offspring dependency on mother's fine-scale movement decisions in L331-333 - "Likewise, the movement decisions of mothers may be highly constrained by small, but independently locomoting offspring (Pontzer et al. 2006)".

Fourth, I am curious about variation in gait length. The authors' analysis of stride frequency are exciting and, I believe, expertly executed. As one additional consideration, though, I wonder if the baboons might be altering gait length as well as stride frequency. Could this be tested using the collars' high resolution GPS data to look at displacement distance per x number of strides? Or would this be over-reaching the spatial accuracy of the GPS readings?

Response 17. The reviewer is raising an important point. However, at this stage, as the reviewer noted, the only reliable measure we can extract from the acceleration signal is stride frequency. Due to the size of GPS sampling error, we cannot extract an unrelated measure of gait length.

Finally, as a minor comment, the authors note that their results are "consistent with the hypothesis that the costs of maintaining cohesion are largely borne by individuals that have the most to gain from group membership" (lines 347-349). My understanding is that they are referencing the fact that certain individuals are disproportionately vulnerable to predation if left alone. Broadly speaking, the benefits of group membership are of course not exclusive to minimized risk of predation (e.g. enhanced reproductive opportunities). Could the authors thus please clarify exactly what they mean by "most to gain" in this statement?

See Response 3.